# PB-LLM: Partially Binarized Large Language Models

**Zhihang Yuan**
Houmo AI

**Zhen Dong**
UC Berkeley

## Abstract

This paper explores network binarization, a radical form of quantization, compressing model weights to a single bit, specifically for Large Language Models (LLMs) compression. Due to previous binarization methods collapsing LLMs, we propose a novel approach, Partially-Binarized LLM (`PB-LLM`), which can achieve extreme low-bit quantization while maintaining the linguistic reasoning capacity of quantized LLMs. Specifically, our exploration first uncovers the ineffectiveness of naïve applications of existing binarization algorithms and highlights the imperative role of salient weights in achieving low-bit quantization. Thus, `PB-LLM` filters a small ratio of salient weights during binarization, allocating them to higher-bit storage, *i.e.,* partially-binarization. `PB-LLM` is extended to recover the capacities of quantized LMMs, by analyzing from the perspective of post-training quantization (PTQ) and quantization-aware training (QAT). Under PTQ, combining the concepts from GPTQ, we reconstruct the binarized weight matrix guided by the Hessian matrix and successfully recover the reasoning capacity of `PB-LLM` in low-bit. Under QAT, we freeze the salient weights during training, explore the derivation of optimal scaling factors crucial for minimizing the quantization error, and propose a scaling mechanism based on this derived scaling strategy for residual binarized weights. Those explorations and the developed methodologies significantly contribute to rejuvenating the performance of low-bit quantized LLMs and present substantial advancements in the field of network binarization for LLMs. The code is available at PB-LLM.

## 1 Introduction

Recently, large language models (LLMs) have gained significant traction in artificial intelligence. It can be attributed to the success of models such as ChatGPT [Brown et al., 2020, Ouyang et al., 2022]. Following its lead, other LLMs such as OPT [Zhang et al., 2022], BLOOM [Scao et al., 2022], and LLaMA [Touvron et al., 2023] have emerged, proving that an increase in model size typically results in enhanced capabilities. As a result, models with tens to hundreds of billions of parameters have become the norm. However, their vast size poses considerable deployment challenges on memory-constrained devices. A model such as the LLAMA-65B (with 65 billion parameters) requires at least 130GB of memory for inference - a number that often exceeds the capacity of a single GPU or server.

Many methods have been proposed to reduce the memory consumption of LLMs [Yuan et al., 2024]. Those methods can be categorized into weight quantization [Dettmers et al., 2022], network pruning [Frantar and Alistarh, 2023], and low-rank factorization [Zhang et al., 2023]. Among these compression paradigms, weight quantization is particularly prominent and widely adopted for LLMs. Since it preserves the original model architecture and leverages well-trained LLMs' full-precision checkpoints, the compression process is greatly simplified [Zhu et al., 2023]. However, state-of-the-art LLM quantization methods show a marked decline in quality beyond 4 bits [Liu et al., 2023a].

More aggressive compression methods are required to push the LLM quantization into the lower bit range. The network binarization technique stands out, reducing the bit-width of weights to just one bit [Helwegen et al., 2019, Rusci et al., 2020, Qin et al., 2020a; 2023]. The binarized models take little storage and memory, and accelerate the inference by efficient bitwise operations. Compared to other aggressive compression technologies like high-sparsity pruning, network binarization has potent topological generics, as it only applies to parameters. Binarization is widely studied in

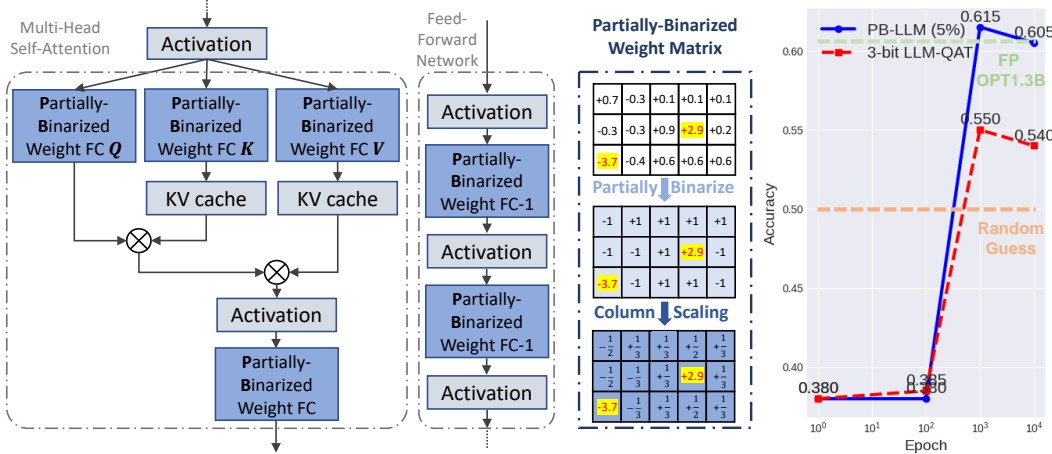

(a) One basic block of the Partially-Binarized LLM.   (b) Performance on BoolQ.

Figure 1: **(a)** We introduce Partially-Binarized Large Language Model (PB-LLM), where a small subset of the weights of the LLM are frozen and preserved with higher bit precision, while the remaining weights are binarized utilizing an optimal scaling factor strategy; **(b)** By using PB-LLM, an extremely low-bit LLM can be acquired efficiently (*i.e.,* quantization-aware training converges quickly) while maintaining its language reasoning capabilities.

academic research as a standalone compression technique, rather than simply a 1-bit specialization of quantization. Some SoTA binarization algorithms have even achieved full-precision performance on large-scale tasks, *e.g.,* ReActNet [Liu et al., 2020a] for ImageNet classification [Deng et al., 2009]. It is theoretically possible to significantly lower the LLM quantization if we generalize the idea of binarizing the weights of LLMs.

In this paper, we explore network binarization specifically for LLM quantization and propose Partially-binarized LLMs (abbreviated as PB-LLM). This methodology aims to achieve extreme quantization to the lowest possible bit, while maintaining the language reasoning capacity inherent in LLMs. The explorations indicate that simple adaptations of existing binarization algorithms do not work well for LLM quantization. As a result of this realization, attention is directed towards the salient-weight property of LLM quantization. In order to achieve the desired extreme low-bit quantization, salient weights must be fully exploited. We investigate the salient weights in aspects of their detection criteria and granularity, as well as the storage costs. Then, we propose the partially binarized matrix, storing the salient weights in higher bits. After establishing the foundation of PB-LLM, the exploration extends to regain the lost reasoning capacity of the quantized LLMs, under the frameworks of post-training quantization (PTQ) and quantization-aware training (QAT). In the view of PTQ, inspired by the concepts of GPTQ [Frantar et al., 2022], we reconstruct the PB-LLM matrix guided by the Hessian matrix and successfully recover the reasoning capacity of PB-LLM in low-bit. In the view of QAT, salient weights are frozen throughout the binarization process for efficient training. In addition, from the perspective of quantization error minimization, we explore how binarized LLMs should be scaled based on the ideal scaling factor. We scale the binarized weight based on the derived scaling strategy shown in Fig. 1a. Low-bit quantized LLMs can significantly improve their performance with such explorations. Benefited from explorations of PTQ and QAT, PB-LLM can efficiently obtain an extremely low-bit LLM with comparable reasoning capacity (see Fig. 1b). The methodologies applied and the insights gained within this study stand to contribute substantially to the advancement of knowledge and development in the field of network binarization for LLMs.

## 2 RELATED WORK

### 2.1 NETWORK BINARIZATION.

Binarization uses the sign function to binarize weights and activations to $\pm 1$. To eliminate the vanishing gradient issue caused by the sign function in the binarization, the straight-through estimator (STE) [Bengio et al., 2013] is utilized for the network backpropagation. Based on this archetype, copious studies contribute to improving the performance of BNNs. Binarization techniques can

be broadly classified into three categories: the enhancement of training objectives, the reduction of gradient mismatch, and the minimization of quantization errors [Qin et al., 2020b; 2023, Yuan and Agaian, 2023]. To illustrate: *Gradient Mismatch:* Liu et al. [2020b] introduce double residual connections paired with full-precision downsampling layers. This approach addresses the gradient vanishing problem that arises due to binarization. *Training Objectives:* Martinez et al. [2020], Shang et al. [2022a;b; 2021] focus on optimizing the loss function during training. They suggest aligning the spatial attention maps derived from both binary and real-valued convolutions. *Quantization Error Minimization:* Rastegari et al. [2016] identify that the disparity in quantization between full-precision and binarized weights can impede the representational abilities of BNNs. As a solution, they introduce a scaling factor—determined by the L1 norm—for both weights and activation functions.

While binarization has proven successful in computer vision, its exploration in natural language processing remains limited. Existing methods [Bai et al., 2020, Qin et al., 2022, Liu et al., 2022; 2023b] primarily target smaller language models (*e.g.,* BERT-base [Devlin et al., 2018] with 110M parameters) potentially hindering their generalization to larger ones (*e.g.,* LLAMA-7B [Touvron et al., 2023] with 7B parameters). We investigate binarization for LLMs comprehensively in this paper and propose `PB-LLM`, which is an attempt to compress LLMs using binarization.

## 2.2 LARGE LANGUAGE MODEL QUANTIZATION.

Quantization, a prominent method in model compression, addresses the storage and computational overhead of deep learning models. Recent research efforts successfully apply quantization to compress Large Language Models (LLMs), including Quantization-Aware Training (QAT) and Post-Training Quantization (PTQ).

In the domain of QAT, innovative strategies like LLM-QAT [Liu et al., 2023a] address challenges in acquiring training data for LLMs by leveraging pre-trained models for data-free distillation. Additionally, techniques such as QLORA [Dettmers et al., 2023a] focus on parameter-efficient fine-tuning (PEFT), expediting model compression and inference acceleration. In PTQ, approaches range from quantizing only the weights of LLMs to jointly quantizing both weights and activations. Methods like GPTQ [Frantar et al., 2022] and QuIP [Chee et al., 2023] optimize matrix multiplications and propose novel layer-wise quantization techniques achieving high compression rates. SqueezeLLM [Kim et al., 2023] and SpQR [Dettmers et al., 2023b] identify weights that lead to particularly large quantization errors and subsequently storing them with higher precision to mitigate the accuracy degradation caused by weight quantization. AWQ [Lin et al., 2023] and OWQ [Lee et al., 2023] contend that when quantizing weights, it is crucial to account for the impact of activation outliers on weights. Norm Tweaking [Li et al., 2023] addresses the issue of activation value deviation by training LayerNorm. For activation quantization, ZeroQuant [Yao et al., 2022] proposes a fine-grained quantization method that can be applied to both weights and activations. Methods like SmoothQuant [Xiao et al., 2022] and Outlier Suppression [Wei et al., 2022; 2023] shift the quantization challenge from activations to weights by proposing a mathematically equivalent per-channel scaling transformation. Omni-Quant [Shao et al., 2023] further enhances performance by training the quantization parameters. RPTQ [Yuan et al., 2023a] proposed proposes performance improvement through grouped quantization after clustering similar channels. In this paper, our primary focus lies in the binarization of weights exclusively, employing both PTQ and QAT methodologies.

## 3 PARTIALLY BINARIZING LARGE LANGUAGE MODELS (`PB-LLM`)

In this section, we elaborate on the methodology of Partially Binarizing Large Language Models, named `PB-LLM`. To begin, a review of the foundational framework of binarized neural networks is presented, showcasing its applicability and limitation to LLM quantization. Subsequently, a novel format for the quantized matrix is formulated, specifically tailored for the binarization of LLMs. Taking advantage of the proposed partially-binarized weight matrix, we delve into its potential in the realms of post-training quantization and training-aware training for LLMs, to break the trade-off between bit-width and performance. It is crucial to note that, due to constraints in computational resources, the methodology exploration predominantly utilizes OPT-1.3B [Zhang et al., 2022] to perform the majority of experiments. Given the space constraints, this section primarily focuses on key aspects of the methodology. For detailed discussions, exact result values, and specific implementation details in codes, readers are referred to the supplemental materials.

### 3.1 PRELIMINARY: NETWORK BINARIZATION

To begin with, we briefly review the general concept of network binarization and binarized neural networks (BNNs) in [Courbariaux et al., 2016, Hubara et al., 2016]. As most optimizable quantized structures of LLMs are linear layers (see Fig. 1a) in LLMs, we use a one-layer Perceptron to show the training and inference processes of the BNN. The one-layer neural network is defined as $f(\mathbf{x}) = (\mathbf{W})(\mathbf{a})$, where $\mathbf{a} \in \mathbb{R}^{d_i}$ is the input activation and $\mathbf{W} : \mathbb{R}^{d_i} \longmapsto \mathbb{R}^{d_o}$ stands for the weight matrix, with $d_i$ and $d_o$ representing the sizes of the input and output of the layer, respectively.

The goal of network binarization is to represent floating-point (FP) weights, denoted as $\mathbf{W}_F$, and/or FP activations $\mathbf{a}_F$ as 1-bit (*i.e., ., $\pm 1$) values [Qin et al., 2020b]. Networks utilizing this representation are referred to as BNNs. BNNs diverge from FP neural networks in their forward operations and in the approximation of backward gradients. In the forward propagation, the sign function is used for binarizing FP values of weights:

$$\text{Forward:} \quad \texttt{sign}(x) = \left\{ \begin{array}{ll} +1 & x \geq 0 \\ -1 & x < 0. \end{array} \right. \tag{1}$$

Specifically, in the training process of binarized network, the BNN maintains FP latent weights $\mathbf{W}_F$ for gradient updates, and the updated weight matrix $\mathbf{W}_F$ is binarized into the binary weight matrix $\mathbf{W}_B$ via the binarize function $\texttt{sign}(\cdot)$, *i.e.* $\mathbf{W}_B = \texttt{sign}(\mathbf{W}_F)$. Then the intermediate activation map (full-precision) of this layer is produced by $\mathbf{A}_{F,o} = \mathbf{W}_B \mathbf{A}_{F,i}$. For inference efficiency, BNNs with 1-bit weights significantly reduce the memory cost of inference. Theoretically, BNNs can binarize both weights and activations to 1-bit, providing a 32x compression in memory cost and a 64x acceleration in inference speed, by replacing FP multiplications in conventional floating-point networks with Xnor-Bitcount operations. However, recent studies highlight that the weights of LLMs as the main contributor to memory overhead [Kim et al., 2023], and thus we primarily aim to curtail memory costs. Therefore, in this pivotal exploration of binarized LLMs, our attention is specifically centered on weight binarization, foregoing the simultaneous binarization of weights and activations.

In the backward propagation, the main challenge is that the pervasive `sign` functions are theoretically non-differentiable, and thus extremely destroy the gradient chain in the backward propagation. To address this problem, researchers widely exploit the straight-through estimator (STE) [Bengio et al., 2013] to numerically approximate the derivative of the whole BNN [Qin et al., 2020b], *i.e.,*

$$\text{Backward:} \quad \frac{\partial \mathcal{L}}{\partial x} = \left\{ \begin{array}{ll} \frac{\partial \mathcal{L}}{\partial \texttt{sign}(x)} & |x| \leq 1 \\ 0 & |x| > 1, \end{array} \right. \tag{2}$$

which makes the optimization of BNN accessible.

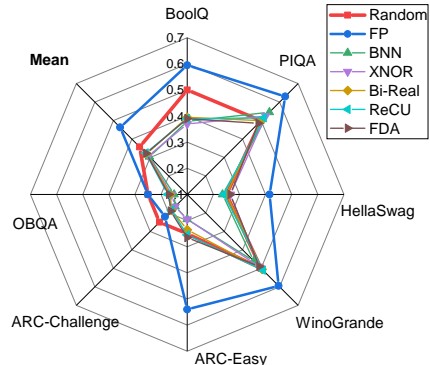

Figure 2: We implement five renowned binarization methods on LLMs and assess the resultant binarized LLMs across seven zero-shot common sense reasoning tasks. Random represents the hypothetical worst baseline, indicating random guesses, while FP stands as the optimal baseline, representing full-precision OPT-1.3B. The exact values corresponding to this radar graph are detailed in the Appendix.

We first investigate the **possibility of implementing binarization to LLM quantization**. Specifically, following the binarization benchmark in BiBench [Qin et al., 2023], we generalize some representative binarization methods into LLM quantization scenarios. BNN [Hubara et al., 2016], XNOR [Rastegari et al., 2016], Bi-Real [Liu et al., 2020b], ReCU [Xu et al., 2021a] and FDA [Xu et al., 2021b] are re-implemented to quantize LLMs, particularly to OPT [Zhang et al., 2022]. Training details are illustrated in the Sec. 4. The results evaluated on seven zero-shot common sense reasoning tasks are shown in Fig. 2. We can see that the LLMs binarized via the existing popular binarization algorithms perform worse than random guesses, showing that the existing binarization methods are not suitable for LLM binarization.

### 3.2 PARTIALLY BINARIZED WEIGHT MATRIX

In the low-bit quantization of Transformers, a significant challenge is managing the salient weights, as they can unnecessarily extend the quantization range [Kovaleva et al., 2021]. Several outlier-aware

LLM compression methods have been explored to tackle this issue [Dettmers et al., 2022, Wei et al., 2022, Kim et al., 2023, Lin et al., 2023, Yuan et al., 2023b]. Notably, SqueezeLLM [Kim et al., 2023] provides a generalized methodology for handling outliers in weight values during 4-bit LLM post-training quantization. Concurrently, AWQ [Lin et al., 2023] demonstrates that preserving only $1\%$ of significant weights can benefit 4-bit LLM quantization. Motivated by existing research, this study also seeks to optimize the treatment of salient weights while binarizing most of weights. We present Partially-Binarized LLMs (`PB-LLM`), a method involving the selective binarization of the LLMs' weight matrix, wherein a minor fraction of weights is kept in high bits for enhanced language capacity.

### 3.2.1 SALIENT WEIGHT: CRITERIA, GRANULARITY, AND COST

Beyond the most straightforward method of choosing salient weights—selecting based on magnitude element-wise—we conduct a thorough investigation into salient weight detection from two perspectives: criteria and granularity. For criteria, we compare Magnitude- and Hessian-based methods, and for granularity, we explore both element-wise and column-wise approaches. In addition, we discuss the cost of storing matrix weights in a mixed-precision manner.

**Criteria: Magnitude vs. Hessian.** Beyond the identification of salient weights through magnitude, alternative criteria have also been examined. The Hessian metric emerges as a crucial factor in LLM quantization, as elucidated in [Dong et al., 2019, Frantar et al., 2022, Frantar and Alistarh, 2023], particularly in relation to post-training quantization for LLMs (details regarding the Hessian criteria for PTQ can be found in Sec. 3.3). However, we observe that the selection of salient weights, whether by magnitude or Hessian, does not significantly impact the efficacy of LLM partial binarization, especially under the framework of QAT. Consequently, magnitude is elected as the preferred criterion for the identification of salient weights in both PTQ and QAT, primarily due to its simplicity and efficacy in distinguishing critical weight components.

**Granularity: Element-wise vs. Column-wise.** Our investigations reveal that adopting a column-wise approach for selecting salient weights has the potential to impair the performance of binarization. Visualization of the salient weights' distribution within the matrix, as depicted in Fig. 3 (where the white dots represent the filtered salient weights), disclosed a random and uniform scattering of these weights. Given the absence of any discernable column-wise pattern in the distribution of salient weights, a column-wise filtration method is deemed unsuitable. This scattered and uniform distribution necessitates an element-wise approach for effective filtration in the binarization process.

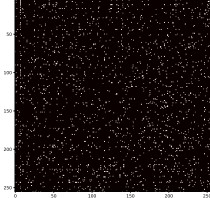

Figure 3: Distribution of $5\%$ salient weight.

**Salient Weight Storing Cost.** The additional overhead for storing the salient weights is acceptable. The overall bit number, $N_{bit}$ must adhere to the following condition:

$$N_{bit} \leq \overbrace{1 * r_{binary}}^{\text{for binary weights}} + \overbrace{N_{salient-bit} * (1 - r_{binary})}^{\text{for salient weights}} + \overbrace{1}^{\text{for index storing, could be optimized}}, \qquad (3)$$

Here, $r_{binary}$ denotes the ratio of the binarized weights, $N_{salient-bit}$ represents the number of bits allocated for storing salient weights (*e.g.,* 8 bits), and the additional 1 bit is allocated for using the bitmap mechanism [Chan and Ioannidis, 1998] for index saving. It's important to note that employing bitmap for index storage is not the most efficient method and can be optimized further using sparse matrix storage methods such as Compressed Sparse Row (CSR) or Compressed Sparse Column (CSC) [Borštnik et al., 2014]; hence the use of $\leq$ instead of $=$ in Eq. 3. The relationship between the ratio of salient weights and the overall bit number is illustrated in Fig. 4, depicting that a lower ratio corresponds to a reduced overall bit number. For example, retaining $10\%$ of weights in 8 bits and binarizing the remaining $90\%$ equates to, at most, a 2.7-bit quantization.

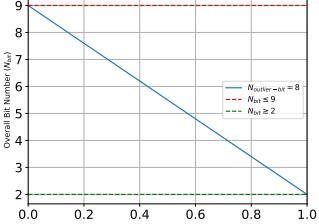

Figure 4: Variation in overall bit number $N_{bit}$ with the ratio of the salient weights $r_{binary}$, where salient weights are stored in 8-bit.

Table 1: Perplexity of C4 on OPT-1.3B quantized with RTN (without GPTQ) and `PB-GPTQ`. Magnitude criteria or Hessian criteria is used for detecting salient weights.

| Salient Fraction | 50% | 20% | 10% | 5% |
|---|---|---|---|---|
| RTN Magnitude | 24.5675 | 5892.0898 | 4889.0385 | 8023.1132 |
| RTN Hessian | 20.2512 | 2109.8522 | 7508.7788 | 6173.1611 |
| `PB-GPTQ` Magnitude | 18.3674 | 46.4093 | 895.0322 | 2880.6157 |
| `PB-GPTQ` Hessian | 17.7567 | 42.1157 | 165.6767 | 528.4877 |
| `PB-GPTQ` Magnitude g=128 | 18.0293 | 57.2164 | 1230.8537 | 2662.7114 |
| `PB-GPTQ` Hessian g=128 | 17.6000 | 45.9811 | 157.8825 | 646.3616 |

## 3.3 POST-TRAINING QUANTIZATION FOR PB-LLMS

After defining the partially-binarized matrix format, the next step is to recover the performance (*i.e.,* the reasoning capacity in the literature of LLMs) of the quantized `PB-LLM`. In this section, we explore the weight binarization with post-training quantization (PTQ) methods. PTQ methods hold a prominent position in the realm of quantization techniques for LLMs due to their ease of implementation. They enable direct quantization of pre-trained LLMs without the need for a training dataset and additional training overhead. Therefore, we first explore the weight binarization within the PTQ framework.

GPTQ [Frantar et al., 2022] is the most efficient and effective method for weight quantization [Zhu et al., 2023], capable of quantizing LLMs to 4-bit or even 2-bit. Therefore, we generalize the idea of GPTQ to the partial-binarization setting. Specifically, GPTQ quantizes the weights in LLM layer-by-layer to minimize the layer-wise quantization error:

$$\arg \min_{\hat{\mathbf{W}}} ||\mathbf{W}\mathbf{X} - \hat{\mathbf{W}}\mathbf{X}||_2^2 \tag{4}$$

GPTQ quantizes a weight $w_q$ to $\hat{w}_q$, calculates the compensation $\delta_{-q}$ for remaining weights $w_{-q}$, and then applies the compensation factor to the remaining weights:

$$\delta_{-q} = \frac{w_q - \hat{w}_q}{[\mathbf{H}^{-1}]_{qq}} \cdot (\mathbf{H}^{-1})_{:,q}, \qquad w_{-q} := w_{-q} + \delta_{-q}, \tag{5}$$

where the $\mathbf{H}$ is the Hessian matrix of the layer-wise quantization error with respect to the weights and $w_q$ is the $q$-th value in flattened weight matrix $\mathbf{W}$. In GPTQ, weights are quantized iteratively and the remaining weights are updated until all weights have been quantized.

We propose to use GPTQ to iteratively binarize the un-salient weights and quantize the salient weights to higher bit, and then apply the compensation to the remaining weights. Specifically, we first detect the salient weights $\mathbf{W}^{sal}$ and un-salient (to-be-binarized) weights $\mathbf{W}^{unsal}$ in the weight matrix $\mathbf{W} = \mathbf{W}^{sal} + \mathbf{W}^{unsal}$. Drawing inspiration from SparseGPT [Frantar and Alistarh, 2023], we calculate the saliency metric, represented as $v_i = w_i^2/[\mathbf{H}^{-1}]_{ii}^2$, for the purpose of detecting salient weights using Hessian criterion. The un-salient weights will be binarized to $\hat{\mathbf{W}}^{unsal}$, and the salient weights will be quantized to higher bit $\hat{\mathbf{W}}^{sal}$. We use asymmetric per-channel quantization for both salient and un-salient weights. For un-salient weight, we use the per-channel mean as zero point and calculate the optimal scaling factor $\alpha$ for the un-salient weights using the method in Sec. 3.4.2. We use MinMax metric to calibrate the scaling factor and zero point for salient weights.

In the quantization process, we iteratively quantize the columns in the weight matrix $\mathbf{W}$. For each column, we binarize the un-salient weights and quantize the salient weights, and then calculate the compensation for remaining weights, and then apply the compensation factor to the remaining columns of weights. This process is repeated until all the weights are quantized. The proposed method is denoted as `PB-GPTQ`. We also explore the fine-grained `PB-GPTQ`, which quantizes the weights in a group-wise manner. Specifically, the weight matrix is split into several groups, each group contains $g$ columns. In each group, we detect the salient weights and un-salient weights, and then calibrate to set the scaling factor and zero point using the weights in this group.

The results are listed in Tab. 1. `PB-GPTQ` is significantly better than RTN. We note that the Hessian-based `PB-GPTQ` exhibits a superior performance compared to the Magnitude criterion `PB-GPTQ`. The group-wise `PB-GPTQ` performs better or worse than the non-group-wise `PB-GPTQ`, but the difference is not significant. Our analysis suggests that the disparity in scaling factors is not the primary determinant of binarization performance; hence, the introduction of group-wise methodology does not yield an enhancement in binarization performance. Subsequently, our next endeavor will involve the application of QAT to reduce the error introduced by weight binarization.

### 3.4 Quantization-aware Training for PB-LLMs

In order to further enhance the reasoning capacity of the Partially-Binarized Large Language Models (`PB-LLM`), we extend our exploration by employing Quantization-aware Training (QAT) to meticulously train the quantized models. Because LLM training is difficult, we desire that `PB-LLM` training could be as efficient as possible. To realize efficient training for `PB-LLM`, we propose the Salient Weights Frozen and Optimal Scaling Factor for Binary Weights, targeting the salient weights and binarized weights, respectively.

#### 3.4.1 Salient Weights Frozen

To leverage the value of pretrained weights, we propose freezing the salient weights, determined by weight magnitude, prior to the weight binarization process. As illustrated in Fig. 1a, we initially filter out a number of weights from a pre-trained weight matrix—*e.g.,* 2% by magnitude—at the beginning of quantization-aware training, maintaining their fixed state throughout the training process. Examination of training efficiency (refer to Fig.5) suggests that these salient weights play a crucial role in LLM capacity. Maintaining the high bit representation of certain weights, aids in the training of quantized LLMs and reduces their optimization difficulty.

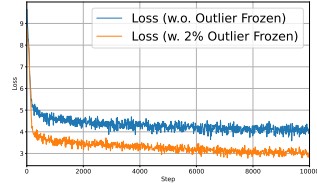

Figure 5: **Training Loss Curves:** When only 2% of weights are retained in their un-binarized state, the training loss converges more swiftly.

#### 3.4.2 Optimal Scaling Factor for Binary Weights.

AWQ [Lin et al., 2023] enhances the weight-only quantization method for LLMs by optimizing scaling factors to mitigate the quantization error of quantized weights. Specifically, AWQ demonstrates that searching for empirically optimal scaling factors proves to be an effective strategy for reducing quantization errors and recovering the performance of the quantized models. Fortunately, in the context of LLM binarization, we have a better choice for scaling the binarized weights. There's no need to search for optimal scaling factors as they can be **analytically derived**. Specifically, we apply a column-wise scaling factor to binarized weights to **reduce the binarization error**, *i.e.,* enforcing $\mathbf{w}_F = \alpha \bar{\mathbf{w}}_B$. The optimal values of scaling factor $\alpha$ for the $\bar{\mathbf{w}}_B \in \{-1, 1\}$ can be calculated by minimizing the L2 error:

$$\alpha^\star = \arg \min_{\alpha \in \mathbb{R}_+} \mathcal{J}(\alpha), \text{ in which } \mathcal{J}(\alpha) = \|\mathbf{w}_F - \alpha \bar{\mathbf{w}}_B\|_2^2 \tag{6}$$

Following XNOR-Net [Rastegari et al., 2016], by expanding the below equation, we have

$$\mathcal{J}(\alpha) = \alpha^2 \bar{\mathbf{w}}_B^T \bar{\mathbf{w}}_B - 2\alpha \mathbf{w}_F^T \bar{\mathbf{w}}_B + \mathbf{w}_F^T \mathbf{w}_F \tag{7}$$

For the vector with $\mathbf{w}_F \in \mathbb{R}^n$ we follow the traditional methods of binarizing weights [Hubara et al., 2016] by taking the sign of real-valued weights:

$$\bar{\mathbf{w}}_B^i = \text{sign}(\mathbf{w}_F^i) = \begin{cases} +1, & \mathbf{w}_F^i \geq 0; \\ -1, & \mathbf{w}_F^i < 0. \end{cases} \tag{8}$$

In that case, $\bar{\mathbf{w}}_B^T \bar{\mathbf{w}}_B = n_{\mathbf{w}_F}$, where $n_{\mathbf{w}_F}$ is number of elements in $\mathbf{w}_F$, and $\alpha^*$ can be solved as:

$$\alpha^* = \frac{\mathbf{w}_F^T \bar{\mathbf{w}}_B}{n_{\mathbf{w}_F}} = \frac{\|\mathbf{w}_F\|_1}{n_{\mathbf{w}_F}}. \tag{9}$$

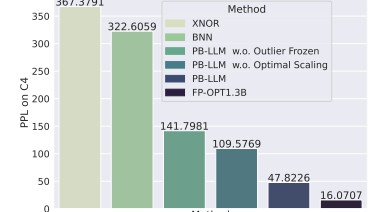

Figure 6: **Perplexity (PPL) on C4:** When 50% of the weights are maintained in their un-binarized state (equivalent to around 5-bit quantization), the untrained `PB-LLM` does not experience a total loss of reasoning capabilities.

A counterintuitive outcome emerges from the incorporation of salient-frozen and optimal-scaling mechanisms: directly deploying those two mechanisms to pre-trained LLM even *without any retraining or fine-tuning*, still results in commendable performance. For instance, applying these techniques to OPT-1.3B with 50% salient weights (see Fig. 6) reveals that the partially-binarized OPT-1.3B retains a small amount of language capacity, corroborating the importance of a small number of salient weights in LLM quantization. Consequently, implementing just these two techniques—Outlier Frozen and Optimal Scaling Factor for Binary Weights—on pre-trained LLMs serves as an efficient starting point for training `PB-LLM`.

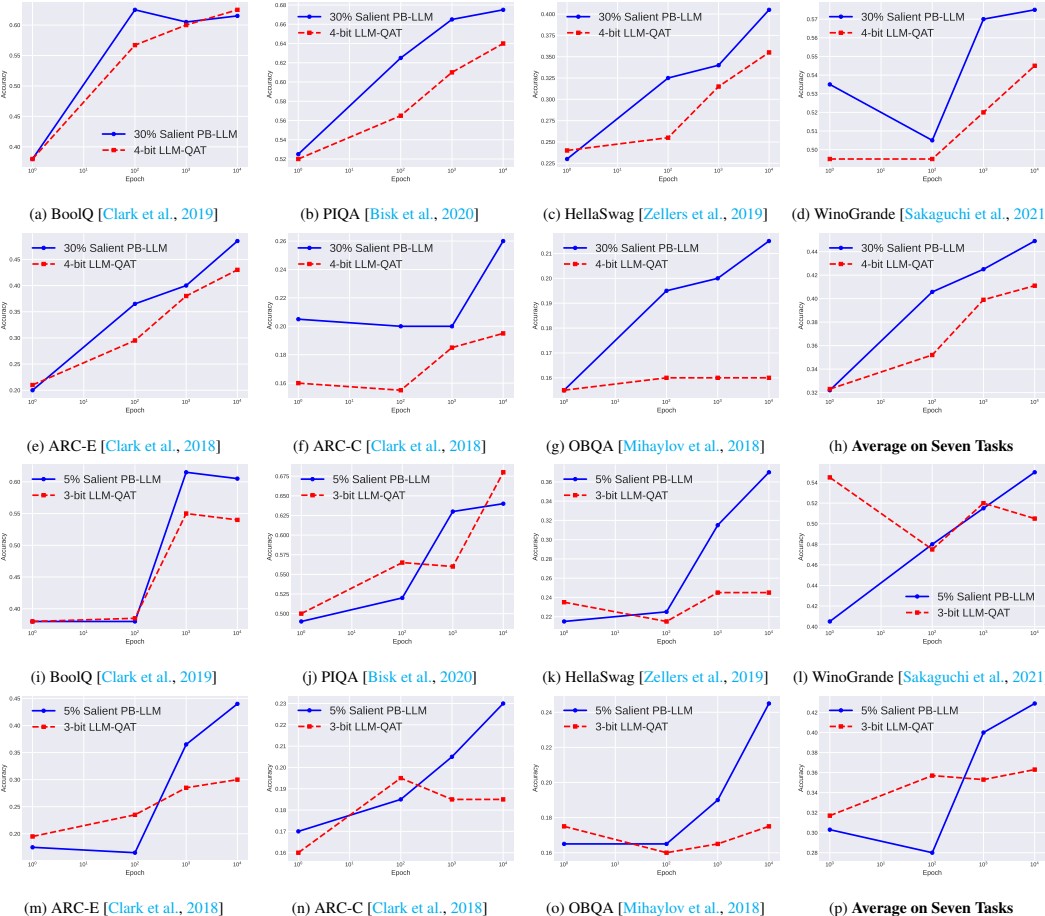

(a) BoolQ [Clark et al., 2019]  (b) PIQA [Bisk et al., 2020]  (c) HellaSwag [Zellers et al., 2019]  (d) WinoGrande [Sakaguchi et al., 2021]

(e) ARC-E [Clark et al., 2018]  (f) ARC-C [Clark et al., 2018]  (g) OBQA [Mihaylov et al., 2018]  (h) **Average on Seven Tasks**

(i) BoolQ [Clark et al., 2019]  (j) PIQA [Bisk et al., 2020]  (k) HellaSwag [Zellers et al., 2019]  (l) WinoGrande [Sakaguchi et al., 2021]

(m) ARC-E [Clark et al., 2018]  (n) ARC-C [Clark et al., 2018]  (o) OBQA [Mihaylov et al., 2018]  (p) **Average on Seven Tasks**

Figure 7: **QAT training results with 30% salient weights `PB-LLM` (upper two lines):** As fine-tuning epochs increase, quantized models swiftly regain their reasoning capacities, demonstrating the resilience and adaptability of `PB-LLM` in sustaining cognitive functionalities within models, despite substantial quantization; **QAT training results with 5% salient weights `PB-LLM` (bottom two lines)**: Existing LLM QAT methods exhibit an absolute failure when subjected to extremely-low bit conditions. In contrast, `PB-LLM` triumphs in restoring the reasoning capacities of low-bit quantized LLMs. This underlines the efficacy of `PB-LLM` in balancing quantization and performance, preserving the essential reasoning abilities of LLMs even under rigorous bit reduction.

Both of the above-proposed mechanisms are very effective when used during quantization-aware training of `PB-LLM`. The consequential outcomes are delineated in Figs.7a-7p. Observations from the presented results elucidate that optimizing using the partially-binarized quantization format is notably more straightforward compared to single-bit quantization. This empirical evidence corroborates the discussion regarding the rapid convergence property found in Sec.3.4.1, highlighting the efficacy and adaptability of our proposed methodology in optimizing LLMs within the constraints of partial binarization. From the perspective of QAT, `PB-LLM` emerges as more efficient in training compared to existing LLM QAT methods. For instance, while models like LLM-QAT [Liu et al., 2023a] necessitate up to 100K iterations for adequate training, `PB-LLM` remarkably achieves recovery of the performance of quantized LLMs in merely around 1-10K iterations.

# 4 EXPERIMENTS

Besides the exploration with OPT-1.3B in Sec. 3, we assess the effectiveness of `PB-LLM` by conducting experiments on LLaMA-7B [Touvron et al., 2023] and presenting results on various tasks.

## 4.1 EXPERIMENTAL SETUP

**Dataset.** In this study, the `PB-LLM` is trained using the RedPajama-simple-1B dataset, as the dataset for LLaMA training is not openly accessible. This dataset, RedPajama-1T, is structured to closely resemble the LLaMa paper and serves as a transparent, open-source alternative to LLM training

Table 2: Zero-shot performance on Common Sense Reasoning tasks within a 4-bit setting. Reported results of previous works are documented in their papers. `PB-LLM` 30% denotes the preservation of 30% salient weights, and `PB-LLM` 10% implies the preservation of 10% salient weights.

| Method | BoolQ | PIQA | HellaSwag | WinoGrande | ARC-E | ARC-C | OBQA | Avg |
|---|---|---|---|---|---|---|---|---|
| FP LLaMA-7B | 76.8 | 79.3 | 76.1 | 70.0 | 73.0 | 48.0 | 57.6 | 68.7 |
| RTN | 71.2 | 77.3 | 72.7 | 66.9 | 68.8 | 46.4 | 52.8 | 65.2 |
| SmoothQuant | 67.7 | 76.0 | 69.4 | 66.7 | 66.9 | 43.0 | 50.6 | 63.0 |
| LLM-QAT | 75.5 | 78.3 | 74.0 | 69.0 | 70.0 | 45.0 | 55.4 | 66.6 |
| PB-GPTQ 10% | 62.3 | 55.9 | 27.7 | 49.3 | 29.3 | 20.1 | 10.6 | 36.5 |
| PB-GPTQ 30% | 73.5 | 74.9 | 47.5 | 64.9 | 61.3 | 32.4 | 25.2 | 54.2 |
| PB-LLM 10% | 68.9 | 67.8 | 68.1 | 67.4 | 58.7 | 42.9 | 50.6 | 60.6 |
| PB-LLM 30% | 75.7 | 78.0 | 74.3 | 69.7 | 69.0 | 45.6 | 55.8 | 66.9 |

dataset. It amalgamates data from diverse sources including Commoncrawl, C4, GitHub, Wikipedia, Gutenberg Books3, ArXiv, and Stackexchange. RedPajama-simple-1B, representing a 0.1% subset of RedPajama-1T, is substantially smaller than the typical datasets used for training other LLMs, making it a convenient choice for our experiments.

**Training Details.** In the training process of our quantized network, we commence with a pre-trained model for initialization. The optimization of the model is facilitated through the AdamW optimizer [Loshchilov and Hutter, 2017], applied with zero weight decay. We assign a batch size of 1 to each GPU and implement a learning rate of 2e-5, adhering to a cosine learning rate decay strategy. We only fine-tune our `PB-LLM` for 10K iterations.

**Evaluated Tasks.** To eliminate the variance of evaluated performance, we evaluate the binarized LLMs on seven zero-shot common sense reasoning tasks, *i.e.,* BoolQ [Clark et al., 2019], PIQA [Bisk et al., 2020], HellaSwag [Zellers et al., 2019], WinoGrande [Sakaguchi et al., 2021], ARC-Easy, ARC-Challenge [Clark et al., 2018], OBQA [Mihaylov et al., 2018]. We also along eavuled the quantized moelds' perplexity scores on WikiText2 [Merity et al., 2016] and C4 [Raffel et al., 2020].

## 4.2 RESULTS ON LLAMA

Experiments were conducted on LLaMA-7B. The results of employing `PB-GPTQ` and `PB-LLM` are illustrated in Tabs. 2 and 3. When employing PTQ, `PB-GPTQ` exhibited commendable performance, particularly when the salient weight exceeded 30%. Nevertheless, a noteworthy decline in the performance of the quantized network was observed when the salient weight was reduced to 10%. On the other hand, employing QAT resulted in a notable improve-

Table 3: Perplexity of C4, wikitext2 and PTB on LLaMA-7b quantized with PTQ methods.

|  | C4 | WIKI | PTB |
|---|---|---|---|
| FP | 7.3435 | 5.6770 | 41.1509 |
| GPTQ 4b | 8.6977 | 8.1368 | 57.9951 |
| SparseGPT 50% | 15.5949 | 12.829483 | 505.1396 |
| PB-GPTQ 50% | 8.1466 | 6.3089 | 54.8674 |
| PB-GPTQ 20% | 20.6057 | 17.1929 | 280.4353 |
| PB-GPTQ 10% | 72.1115 | 85.7838 | 708.4120 |
| PB-GPTQ 5% | 401.6475 | 619.1054 | 1687.1815 |

ment in the performance. A comparison within a 4-bit quantization setting between `PB-LLM` 30% and LLM-QAT in Tab. 2 reveals superior performance by our method. It is notable that `PB-LLM` is only fine-tuned for 10K iterations, whereas LLM-QAT underwent 100K iterations of training, showing its fast convergence property (refer to Sec. 3.4). The results under `PB-LLM` 10% represent the outcomes of `PB-LLM` where 10% of salient weights are preserved. This demonstrates the potential for advancing LLM quantization towards a fully 1-bit state.

## 5 CONCLUSION

In conclusion, this work is the first to implement network binarization for LLM quantification, introducing the novel Partially-binarized LLM (`PB-LLM`) methodology. This approach is meticulously designed to maintain linguistic reasoning capabilities of LLMs, even under extreme low-bit quantization. The research unearthed the significant role of salient weights in achieving extreme quantization and proposed innovative strategies like optimal scaling for effective binarization. This framework is extended to recover the capacities of quantized LMMs, by analyzing from the perspective of post-training quantization (PTQ) and quantization-aware training (QAT). The methodology is a significant stride in the realm of network binarization for LLMs.

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

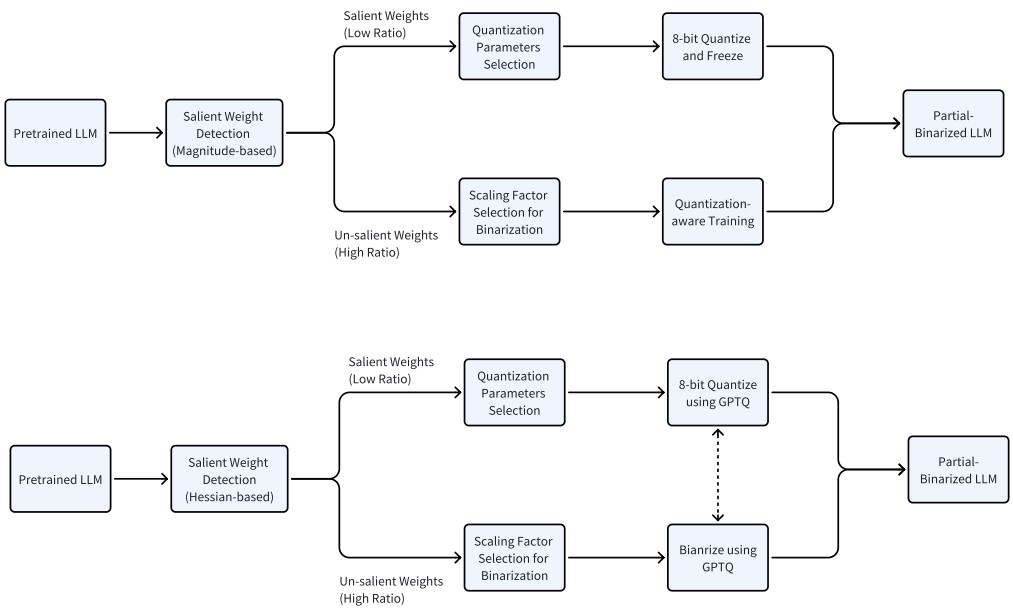

Figure 8: Flowchart of `PB-LLM`.

# A  SUPPLEMENTAL MATERIALS

## A.1  EXISITING BINARIZATION METHODS ON LLM QUANTIZATION

| Method | BoolQ | PIQA | HellaSwag | WinoGrande | ARC-Easy | ARC-Challenge | OBQA | Mean |
|---|---|---|---|---|---|---|---|---|
| Random Performance | 0.5 | 0.5 | 0.25 | 0.5 | 0.25 | 0.25 | 0.25 | 0.36 |
| FP | 0.595 | 0.63 | 0.415 | 0.595 | 0.54 | 0.22 | 0.25 | 0.46 |
| BNN | 0.38 | 0.545 | 0.235 | 0.46 | 0.195 | 0.165 | 0.15 | 0.30 |
| XNOR | 0.37 | 0.525 | 0.265 | 0.49 | 0.195 | 0.165 | 0.16 | 0.31 |
| Bi-Real | 0.395 | 0.5 | 0.25 | 0.505 | 0.235 | 0.185 | 0.165 | 0.32 |
| ReCU | 0.39 | 0.515 | 0.24 | 0.51 | 0.255 | 0.185 | 0.175 | 0.32 |
| FDA | 0.39 | 0.485 | 0.265 | 0.49 | 0.265 | 0.19 | 0.17 | 0.32 |

Table 4: Table corresponds to Figure 2 in the main paper: We implement five renowned binarization methods on LLMs and assess the resultant binarized LLMs across seven zero-shot common sense reasoning tasks.

We first investigate the possibility of implementing binarization to LLM quantization. Specifically, following the binarization benchmark in BiBench [Qin et al., 2023], we generalize some representative binarization methods into LLM quantization scenarios. BNN [Hubara et al., 2016], XNOR [Rastegari et al., 2016], Bi-Real [Liu et al., 2020b], ReCU [Xu et al., 2021a] and FDA [Xu et al., 2021b] are re-implemented to quantize LLMs, particularly to OPT [Zhang et al., 2022]. Training details are illustrated in the Sec. 4. The results evaluated on seven zero-shot common sense reasoning tasks are shown in the above table. We can see that the LLMs binarized via the existing popular binarization algorithms perform worse than random guesses, showing that the existing binarization methods are not suitable for LLM binarization.

## A.2  CODES

Codes can be found anomalously in PB-LLM.

| Criteria | WIKI | PTB | C4 |
|----------|-------|-------|-------|
| Hessian | 37.07 | 47.07 | 38.76 |
| Magnitude | 37.50 | 44.42 | 38.64 |

Table 5: Salient Weight Selection

## A.3 FLOWCHART OF PB-LLM

## A.4 MORE EXPERIMENTS ON SALIENT WEIGHT SELECTION

In our experimental analysis, we employed the OPT-1.3b network and conducted PB-LLM (QAT) quantization under both Hessian and Magnitude criteria, targeting a 10% salient weight. The performance results, measured in perplexity (PPL) are in Tbale 5. We observed that the PPL values were comparable between the two criteria. However, it's worth noting that the Hessian criteria involve an additional calibration step (collecting calibration data to compute the Hessian), adding complexity to the process. In contrast, the Magnitude criteria offer a simpler approach, eliminating the need for this extra calibration step. Given the similar PPL results and the desire to streamline the quantization process in QAT, we opted for the Magnitude Metric to identify salient weights during QAT. This decision was motivated by the goal of simplifying the process while maintaining comparable performance.

