# OpenReview forum: "PB-LLM: Partially Binarized Large Language Models"
_ICLR.cc/2024/Conference — ICLR 2024 poster_

### Official Review · Reviewer_JdJY · 2023-10-31

**Soundness:** 3 good
**Presentation:** 3 good
**Contribution:** 3 good
**Rating:** 8
**Confidence:** 4

**Summary:**

This paper introduces PB-LLM, an extreme quantization method that partially binarizes model parameters based on their relevance to linguistic reasoning tasks. PB-LLM selectively identifies and designates a small fraction of salient weights for binarization, assigning them to higher-bit storage, and essentially implementing partial binarization. PB-LLM is extended to recover quantized LMMs' capabilities through post-training quantization (PTQ) and quantization-aware training (QAT). In PTQ, the Hessian matrix guides the reconstruction of binarized weights to restore reasoning capacity at low bits. In QAT, we freeze salient weights during training, derive optimal scaling factors to minimize quantization errors and propose a scaling mechanism for enhanced residual binarized weights. The result shows that PB-LLM achieves competitive results without losing performance gain.

**Strengths:**

+ Presents a practical approach that analytically identifies non-salient weights and applies binarization selectively for large models.
+ Achieves LLM binarization without sacrificing performance gains.

**Weaknesses:**

- The proposed approach section is comprehensive, but its complexity makes it challenging to navigate and comprehend throughout the entire section.
- The analysis of evaluation is limited to a single task. It would be valuable to explore the potential limitations of PB-LLM in achieving comparable performance across various tasks.

**Questions:**

1. The readability of the paper can be improved by including a flowchart or block diagram of the proposed method by illustrating different stages of transforming a PB-LLM model.
2. The paper reports primarily the accuracy to evaluate the proposed approach. The addition of the final model size after partial binarization can further solidify the claim.

---

> ### Author Response · Authors · 2023-11-19
> **Response to Reviewer JdJY**
>
> ## To Weakness 1 & Question 1: Enhancing the Clarity
>
> > Weakness 1: The proposed approach section is comprehensive, but its complexity makes it challenging to navigate and comprehend throughout the entire section. Question 1: The readability of the paper can be improved by including a flowchart or block diagram of the proposed method by illustrating different stages of transforming a PB-LLM model.
>
> Thank you for your valuable feedback.
>
> To enhance clarity, we have succinctly summarized our PB-LLM approach and included a conceptual illustration in Fig. 1 (a). PB-LLM begins with a pre-trained, full-precision LLM and partially binarizes the weight matrix while preserving salient weights in higher bits. The process is designed to recuperate the quantized LLM's reasoning capacity, employing both post-training quantization (PTQ) and quantization-aware training (QAT) frameworks. For PTQ recovery (GPTQ-PB), we reconstruct the partially-binarized matrix, guided by Hessian metrics. For QAT recovery (PB-LLM), salient weights are frozen during binarization to streamline the training process.
>
> Additionally, we have incorporated your suggestion by adding a detailed flowchart in the Appendix. This flowchart delineates the stages of transforming a PB-LLM model under the frameworks of PTQ and QAT, intending to improve readability and comprehension.   The revisions are highlighted in red for easy identification and reference. We integrate this diagram into the main body of the paper for the camera-ready version.
>
> ## To Weakness 2: More Evaluation on MMLU 57  Tasks
>
> > Weakness 2: The analysis of evaluation is limited to a single task. It would be valuable to explore the potential limitations of PB-LLM in achieving comparable performance across various tasks.
>
> Thank you for your insightful suggestion. To address your concern regarding the evaluation's scope, we extended our analysis to include the few-shot MMLU benchmarks. Specifically, we applied PB-LLM to the LLaMA-7B model, preserving 30% of salient weights, which is analogous to a 4-bit quantization as per LLM-AQT. We then evaluated this quantized LLaMA-7B across 57 MMLU tasks.
>
> Below are the comparative results of our method against established post-training quantization (PTQ) and quantization-aware training (QAT) methods, such as SmoothQuant and LLM-QAT:
>
> | Method    | Humanities | STEM | Social Sciences | Other | Average |
> |-----------|----------|-----------|----------------|---------|---------|
> | Full-precision LLaMA-7B   | 33.5 | 30.6 |  38.4 | 39.1 | 35.2 |
> | SmoothQuant (PTQ)  | 26.9 | 28.6 | 29.6 | 32.0 | 29.0 |
> | LLM-QAT (QAT)  | 30.3 | 28.1 | 30.3 | 34.5 | 30.8 |
> | PB-LLM (30%)  | 30.5 | 29.0 | 30.2 | 34.6 | 31.2 |
>
>
> It is important to note that we have not conducted exhaustive hyper-parameter tuning in this new setting. Nonetheless, these results are indicative of our method’s generalization ability across various tasks, even without fine-tuned hyper-parameters. Meanwhile, PB-LLM is training efficient (refer to Response to Reviewer 92rD)
>
> ## To Question 2: Model Storage Saving
>
> > Question 2: The addition of the final model size after partial binarization can further solidify the claim.
>
> We appreciate your suggestion to detail the model size savings achieved through our method. Our PB-LLM approach, which involves selective binarization of the LLMs' weight matrix, indeed leads to significant storage savings. This is achieved by retaining a portion of the weights in high bits to bolster linguistic capacity, without substantially increasing the overall model size.
>
> The overall bit number, $N_{bit}$, of our model is calculated using the formula: $N_{bit} \leq 1 \times r_{binary} + N_{salient-bit} \times (1 - r_{binary}) + 1$. In this formula, $r_{binary}$ represents the ratio of weights that are binarized, and $N_{salient-bit}$ signifies the bit count allocated for salient weights (e.g., 8 bits), with an additional bit dedicated to a bitmap index mechanism. As an illustrative example, if 10% of the weights are designated as salient and stored in 8 bits, the overall bit requirement would be at most 1$\times$ 90% + 8$\times$ 10% + 1= 2.7 bits. This implies that the storage requirement for our quantized model is just 16.8% of what would be needed for a full 16-bit model (2.7/16 = 16.8%).

---

> > ### Author Response · Authors · 2023-11-21
> > **2 DAYS REMAINING: We would like to learn Reviewer’s opinion and address any remaining concerns.**
> >
> > Dear Reviewer JdJY,
> >
> > As there are two days remaining for the discussion period, we would kindly like to inquire if you would get a chance to review our response and if there are any remaining questions we can address.
> >
> > Your insights, both the constructive suggestions and areas of contention, have been crucial for us. We have done our best to address each point and clarify any misunderstandings. We are truly keen to have a constructive dialogue with you to refine our work further.
> >
> > Best regards,
> >
> > Authors.

---

> > > ### Comment · Reviewer_JdJY · 2023-11-22
> > >
> > > The authors have addressed all the issues that I raised, and I'd be happy to increase my rating.

---

> > > > ### Author Response · Authors · 2023-11-23
> > > > **Thanks for Raising the Score to 8, Reviewer JdJY**
> > > >
> > > > Dear Reviewer JdJY,
> > > >
> > > > Thank you for your acknowledgment and insightful feedback. We are grateful for your recognition of our efforts in addressing the concerns you raised, and deeply appreciate the expertise you have brought to the refinement of our work. Your suggestions have been instrumental in enhancing our manuscript. We greatly appreciate that you raised the original rating  borderline accept (6) to accept (8). We have included the supplemented results in the updated paper.
> > > >
> > > > Thank you very much,
> > > >
> > > > Authors

---

### Official Review · Reviewer_9nFP · 2023-11-03

**Soundness:** 2 fair
**Presentation:** 3 good
**Contribution:** 2 fair
**Rating:** 6
**Confidence:** 3

**Summary:**

This paper proposes a Partially-Binarized LLM (PB-LLM), which leverages a PB mechanism to binarize LLMs for more efficient GPU memory usage. The main motivation behind PB is the recognition that a fraction of 'salient weights' exists within the LLM weights, which are essential and restrict full binarization. PB-LLM preserves the precision of these salient weights while focusing on binarizing the non-salient weights. The paper also investigates the integration of PB-LLM with post-training quantization and quantization-aware training schemes and establishes a connection between PB-LLM and GPTQ. Experimental results are presented to demonstrate the effectiveness of PB-LLM.

**Strengths:**

- The paper is well-written and well-motivated.
- The proposed PB-LLM scheme is easy to follow and straightforward to understand.
- Exploring to improve the memory and/or storage efficiency using quantization (and algorithmic approximation in general) of LLMs is a promising research direction.

**Weaknesses:**

- Only Llama 7B is studied as the LLM for PB-LLM and all other baselines. Thus, it's not clear how the PB-LLM method performs on larger-scale models.
- Only pre-trained base models are experimented with those models, however, are usually not deployed directly as applications.
- The LLM quantization scheme is motivated using an angle of GPU memory efficiency. However, the actual GPU memory usage before and after binarization/quantization is not studied in this paper.

**Questions:**

- How does PB-LLM's performance vary among various sizes of Llama models, e.g., 7B-65B?
- How does PB-LLM perform for aligned/instruction fine-tuned models, e.g., Alpaca and/or Vicuna? To what extent will the binarization scheme affect the model's performance say on the Hugging Face leaderboard [1]?
- What is the actual GPU memory saving look like for PB-LLM and all considered baselines?

[1] https://huggingface.co/spaces/HuggingFaceH4/open_llm_leaderboard

---

> ### Author Response · Authors · 2023-11-19
> **Responses to Reviewer 9nFP**
>
> ## To Weakness 1 & Question 1: Extensive Experiments on LLaMA-13B and LLaMA-30B.
>
> > Weakness 1: Only Llama 7B is studied as the LLM for PB-LLM and all other baselines. Thus, it's not clear how the PB-LLM method performs on larger-scale models. Question 1: How does PB-LLM's performance vary among various sizes of Llama models, e.g., 7B-65B?
>
> Thank you for acknowledging the various aspects of our work and for your valuable recommendation. In response, we have extended our experimentation to include larger LLaMA models, specifically LLaMA-13B and LLaMA-30B, to gauge the scalability of our PB-LLM method. Unfortunately, due to computational resource constraints, we were unable to conduct experiments on the LLaMA-65B model, as our available resources – an 8 A6000s server – represent the upper limit for our academic lab's resources.
>
> Below are the results showcasing PB-LLM's performance across these additional LLaMA model sizes:
> | Method    | BoolQ | PIQA |  HellaSwag | WinoGrande | ARC-E | ARC-C | OBQA | Avg |
> |-----|------|----------|-----|------|--------|--------|---------|---------|
> | LLaMA-13B Full-precision | 78.1 | 80.0 | 79.2 | 73.6 | 74.5 | 52.6 | 55.0 | 70.5|
> | RTN     | 76.8 | 79.1 | 76.3 | 70.5 | 72.6 | 49.8 | 56.6 | 68.8|
> | LLM-QAT | 77.7 | 79.4 | 77.7 | 71.5 | 72.8 | 52.0 | 53.8 | 69.2|
> | 30% PB-LLM | 78.9 | 79.9 | 78.5 | 72.0 | 72.9 | 51.3 | 54.1| 69.7|
>
> | Method    | BoolQ | PIQA |  HellaSwag | WinoGrande | ARC-E | ARC-C | OBQA | Avg |
> |-----|------|----------|-----|------|--------|--------|---------|---------|
> | LLaMA-30B Full-precision | 83.2 | 82.1 |82.9 | 75.6 | 80 | 58 | 59.3 | 74.4|
> | RTN     | 80.8 | 80.1 | 81.6 | 75.8 | 79.3 | 55.8 | 57.2 | 72.9|
> | LLM-QAT | 81.8 | 81.0 |81.8 | 75.1 | 79.4 | 56.8 | 54.9 | 72.9|
> | 30% PB-LLM | 82.3 | 79.9 | 81.8 | 76.5 | 79.8 | 56.4 | 56.0 | 73.3|
>
> It is important to note that exhaustive hyper-parameter tuning has not been conducted in this expanded setting. However, these results, even without finely tuned hyper-parameters, indicate the generalization capability of our method across various model scales. In the meantime, our method is more training efficient (refer to Responses to Reviewer 92rD).
>
> ## To Weakness 2 & Question 2: Extensive Experiments on Vicuna-7B.
>
> > Weakness 1: Only pre-trained base models are experimented with those models, however, are usually not deployed directly as applications. Question 1: Only pre-trained base models are experimented with those models, however, are usually not deployed directly as applications.
>
> Thanks for your suggestion. We are testing the performance of PB-LLM on Vicuna-7B (we use this checkpoint from huggingface: https://huggingface.co/AlekseyKorshuk/vicuna-7b). The results will come out in 8 hours.
> ***
> **Nov/19/2023 Results Update:**
>
> We acknowledge the importance of validating our method on models that extend beyond pre-trained base models, as these are more representative of real-world applications. To address this, we have conducted additional experiments using Vicuna-7B. The results of these experiments are as follows:
>
> | Method    | Humanities | STEM | Social Sciences | Other | Average |
> |-----------|----------|-----------|----------------|---------|---------|
> | Full-precision Vicuna-7B   | 47.6 | 40.4 | 55.9 | 54.5 | 49.6 |
> | LLM-QAT (QAT)  | 40.1 | 31.2 | 46.2 | 47.8 | 41.4|
> | PB-LLM 30% Salient Weight  | 41.5 | 34.1 | 50.7 | 51.1 | 45.6|
>
> Based on our **codebase provided in the supplementary materials**, we applied PB-LLM to quantize Vicuna-7B. The results, showcased above, are from the MMLU benchmarks. For comparison, we also quantized Vicuna-7B using LLM-QAT, limiting the training to 10K iterations, identical to the training duration for our PB-LLM method, due to resource constraints.
>
> These results demonstrate that even when applied to fine-tuned models like Vicuna, our PB-LLM method maintains its efficacy, showcasing superior performance compared to LLM-QAT under similar training conditions. This illustrates the versatility and robustness of our approach, extending its applicability to a broader range of LLM deployment scenarios.

---

> ### Author Response · Authors · 2023-11-20
> **Responses to Reviewer 9nFP (II)**
>
> ## To Weakness 3 & Question 3: actual GPU memory saving.
>
> >  To Weakness 3: The LLM quantization scheme is motivated using an angle of GPU memory efficiency. However, the actual GPU memory usage before and after binarization/quantization is not studied in this paper.  Question 3: What is the actual GPU memory saving look like for PB-LLM and all considered baselines?
>
> We acknowledge the importance of understanding the actual GPU memory usage before and after binarization/quantization. Unfortunately, our team lacks dedicated GPU implementation experts, which hindered us from providing a comprehensive analysis in the main body of the paper. Realizing real GPU implementation for algorithms often involves a large team.
> To address your concern, we conducted additional experiments focusing on GPU memory savings for PB-LLM. We utilized the CSR (Compressed Sparse Row) sparse format for 8-bit quantized salient weights.The results are summarized below, comparing the memory occupancy of salient weights after various levels of quantization:
>
> | Configuration | PB-LLM 10% | PB-LLM 20%| PB-LLM 30% | 2bit Quant| 4bit Quant|
> |-----------|----------|-----------|----------------|---------|---------|
> | Memory Occupancy Comparing with FP16   | 16.25% | 26.25% |  36.25% | 12.50% | 25.00% |
>
>
> Notably, PB-LLM with 10% salient weights exhibits memory overhead similar to 2-bit quantization, while PB-LLM with 20% salient weights shows memory overhead comparable to 4-bit quantization. Note that some advanced salient location-saving techniques, such as lead zero, can further decrease the ratio. We plan to incorporate these results into the camera-ready version of the paper. Additionally, we recognize the potential for further optimization in sparse storage. Future work may explore enhancements such as leveraging col index and row index stored in the CSR format of salient weights for binarized quantization or employing semi-structured sparse optimization techniques.

---

> > ### Author Response · Authors · 2023-11-21
> > **2 DAYS REMAINING: We would like to learn Reviewer’s opinion and address any remaining concerns.**
> >
> > Dear Reviewer 9nFP,
> >
> > As there are two days remaining for the discussion period, we would kindly like to inquire if you would get a chance to review our response and if there are any remaining questions we can address.
> >
> > Your insights, both the constructive suggestions and areas of contention, have been crucial for us. We have done our best to address each point and clarify any misunderstandings. We are truly keen to have a constructive dialogue with you to refine our work further.
> >
> > Best regards,
> >
> > Authors.

---

> > > ### Author Response · Authors · 2023-11-22
> > > **LESS THAN 1 DAY REMAINING: We would like to learn Reviewer’s opinion and address any remaining concerns.**
> > >
> > > Dear Reviewer 9nFP,
> > >
> > > As there is less than 1 day remaining for the discussion period, we would kindly like to inquire if you would get a chance to review our response and if there are any remaining questions we can address.
> > >
> > > Best regards,
> > >
> > > Authors.

---

> > > > ### Comment · Reviewer_9nFP · 2023-11-22
> > > > **Thanks for the rebuttal**
> > > >
> > > > I thank the authors for carefully addressing all my concerns and for their hard work in adding such thorough experimental results.
> > > >
> > > > My major concerns have been addressed, and I would like to increase my evaluation score from 5 to 6.

---

> > > > > ### Author Response · Authors · 2023-11-22
> > > > > **Thanks for Raising the Score to 6, Reviewer 9nFP**
> > > > >
> > > > > Dear Reviewer 9nFP,
> > > > >
> > > > > Thanks for the acknowledgment and valuable feedback. We're grateful for your recognition of our efforts to address the concerns you raised. Your expertise has significantly contributed to the enhancement of our work. We would also greatly appreciate that you raised the original rating (5) to borderline accept (6). We will include the supplemented results in the final version of the paper.
> > > > >
> > > > > Thank you very much,
> > > > >
> > > > > Authors

---

### Official Review · Reviewer_tXaS · 2023-11-05

**Soundness:** 2 fair
**Presentation:** 3 good
**Contribution:** 3 good
**Rating:** 5
**Confidence:** 5

**Summary:**

This paper presents a methodology termed PB-LLM for the partially weight-only binary quantization for Large Language Models (LLMs). Particularly, it outlines the limitations encountered when applying previous BNN works to LLMs, while proposing new methods from Post Training Quantization (PTQ) and Quantization Aware Training (QAT) for LLM low-bit weight quantization. Within PTQ, the paper utilizes Hessian information to identify element-wise salient weights, upon which 8-bit quantization is applied to preserve reasoning capacity in a pre-trained LLM, while binary (1-bit) quantization is applied to the un-salient weights. After applying mixed-quantization, it employs GPTQ to un-salient weights for further optimization (termed as GPTQ-PB). In QAT, the paper applies QAT during the LLM fine-tuning stage, utilizing 1) optimal scaling factor and 2) salient weight freezing methodology to bring about an efficient yet higher-performing QAT methodology (termed as PB-LLM). The methodologies studied and proposed in this work provide insightful contributions to the field of LLM binarization, showcasing the potential for further advancements in LLM binarization.

**Strengths:**

- The paper provides a well-structured presentation of the preliminaries of binary quantization to the introduction of the proposed method, which made it easy to follow.

- Building on previous research such as AWQ and SparseGPT, this paper proposes a partial binarization technique through salient weight protection based on the hessian information and effectively demonstrates its efficacy in PTQ with ablation study (as shown in Table 1).

- The paper showcases empirical improvements in QAT optimization (higher accuracy with fewer training steps than LLM-QAT) across the CSQA tasks.

**Weaknesses:**

Major Concerns
- Lack of novelty: The authors propose the optimal scaling factor as their primary contribution, but the core idea itself seems to have already been proposed in the previous work. For example, [R1] proposed an optimal ternary function (eq. (3) of [R1]), but it can be trivially reduced to the binary function when the threshold is zero; then the equation seems to be identical to the equation (8) and (9) proposed in this paper.

[R1] Li te tal., Ternary Weight Networks


- Lack of Evaluation Task: This paper evaluates the reasoning capability of LLM only through the accuracy of the CSQA task. In the CSQA task, tasks such as OBQA and ARC challenge were used for OPT-1.3B, where the FP performance did not reach even the random baseline (25%). It raises questions about the suitability of these tasks for demonstrating the effectiveness of fine-tuning, and, hence, the superiority of PB-LLM. To show the effects of fine-tuning more clearly, it would be advisable to carefully select reasoning tasks that are appropriate for the model capacity. Reporting performance not just on CSQA, but also on multi-task accuracy like MMLU would be also beneficial for highlighting PB-LLM's efficacy.

- Inconsistent Salient Weight Methodology between PTQ and QAT: The absence of a consistent methodology for salient weight protection between PTQ and QAT is concerning. While the effectiveness of using Hessian criteria for identifying salient weights in PTQ is demonstrated through performance comparisons, the rationale for using magnitude criteria to identify salient weights in QAT seems to be missing. Understanding the disparity in the approach to salient weight protection across PTQ and QAT is crucial for a holistic appreciation of the proposed method.

- Insufficient evidence on PB-LLM efficiency: To claim that PB-LLM is more efficient in terms of training iteration number compared to LLM-QAT, a more thorough comparison seems necessary. Specifically, it needs to be clear whether the LLM-QAT, being compared with PB-LLM, has been fine-tuned on the same dataset as PB-LLM. Detailed experimental setup information regarding the LLM-QAT is required. Moreover, verification is needed on whether the results through PB-LLM QAT have fully enhanced the reasoning capacity of the pre-trained model. Essentially, it appears that the reasoning accuracy of the target model (OPT-1.3B) obtained through FP fine-tuning should be presented as the upper bound in Figure 7. Additionally, there seems to be a lack of information in Table 2 regarding whether FP LLaMA-7B performance is pre-trained or fine-tuned.


Minor Concerns
- Typo: Sec 3.3 bianrize -> binarize
- Consistent notation should be used in Sec 4.1 -> LLaMA, LLaMa -> LLaMA
- There may be an incorrect reference link in Sec 4.1, "showing its fast convergence property (refer to 3.2)" Should it possibly be corrected to "refer to 3.4?"
- There are spacing issues in the Figure 7 caption, "LLMin" should be "LLM in" and "PM-LLMtriumphs" should be "PB-LLM triumphs".

**Questions:**

1. What is the rationale behind utilizing a Hessian-based approach to identify salient weights in PTQ, while employing a magnitude-based approach to identify salient weights in QAT?

2. Is there a specific reason why only the PB methodology and GPTQ were applied in PTQ? I am curious about the performance of AWQ-PB in comparison.

3. Is there a plan to compare the task-specific fine-tuning in QAT with the 2-bit QAT-KD methodology (QuantGPT [R2], TSLD [R3])?

4. The optimal scaling factor and salient weight freezing seem to primarily aim at reducing harsh quantization errors before QAT. Is there insight into how these methods improve the QAT training process as depicted in Figure 5?

[R2] Tao et al, " Compression of Generative Pre-trained Language Models via Quantization" ACL 2022.
[R3] Kim et al, "Token-scaled logit distillation for ternary weight generative language models" NeurIPS 2023

---

> ### Author Response · Authors · 2023-11-19
> **Responses to Reviewer tXaS (I)**
>
> ## To Weakness 1: Novelty Clarification
>
> > Lack of novelty: The authors propose the optimal scaling factor as their primary contribution, but the core idea itself seems to have already been proposed in the previous work. For example, [R1] proposed an optimal ternary function (eq. (3) of [R1]), but it can be trivially reduced to the binary function when the threshold is zero; then the equation seems to be identical to the equation (8) and (9) proposed in this paper.
> [R1] Li te tal., Ternary Weight Networks
>
> We appreciate the reviewer’s insight and the opportunity to clarify the novelty and significance of our approach within the context of LLM compression.
> Firstly, we wish to emphasize that our paper does not claim undue novelty in the concept of optimal scaling factors, a concept initially popularized by XNOR-Net, which itself has a substantial citation record (a paper in 2016, over 5000 citation). Indeed, as indicated in Section 3.4.2 of our paper, we acknowledge this foundational work. Even Ternary Weight Networks, which you referenced, credits XNOR-Net for its parameter optimization strategy. Our citation of XNOR-Net is, therefore, appropriate and recognizes the original contribution of this seminal work. We believe that the application of the optimal scaling concept in the context of LLMs, as we have done, represents a novel and significant contribution, similar to how Ternary Weight Networks adapted these ideas to their framework.
>
> Secondly, to clarify, our novel contributions are as follows:
>
> -  Pioneering Network Binarization in LLM Quantization: We are the first to introduce and conduct a comprehensive exploration of network binarization specifically for LLM quantization, demonstrating its capability to achieve lower-bit compression.
> - Development of PB-LLM: We propose the Partially-Binarized LLM (PB-LLM), which incorporates a unique salient weights freezing mechanism and applies optimal scaling strategies to preserve the functionality of quantized models.
> - Exploring PTQ and QAT for Performance Recovery: Our work is the first application of post-training binarization in the context of PTQ and leverages QAT to efficiently train PB-LLM, showcasing both as practical and effective strategies for LLM compression.
>
> We trust these points clarify the originality and value of our research, affirming its unique contribution to the field of LLM compression.

---

> ### Author Response · Authors · 2023-11-19
> **Responses to Reviewer tXaS (II)**
>
> ## To Weakness 2: Extensive Evaluation on MMLU
>
> > Lack of Evaluation Task: This paper evaluates the reasoning capability of LLM only through the accuracy of the CSQA task. In the CSQA task, tasks such as OBQA and ARC challenge were used for OPT-1.3B, where the FP performance did not reach even the random baseline (25%). It raises questions about the suitability of these tasks for demonstrating the effectiveness of fine-tuning, and, hence, the superiority of PB-LLM. To show the effects of fine-tuning more clearly, it would be advisable to carefully select reasoning tasks that are appropriate for the model capacity. Reporting performance not just on CSQA, but also on multi-task accuracy like MMLU would be also beneficial for highlighting PB-LLM's efficacy.
>
> Thank you for your valuable feedback. To clarify and address your concerns:
>
> **Choice of Model for Exploration Experiments:**
>
> The primary reason for utilizing OPT-1.3B in our initial experiments was its accessibility and compatibility with our available GPU resources. Using a single A6000, we could efficiently test most modules in the method exploration phase. However, acknowledging the limitations of smaller models like OPT-1.3B, we have expanded our methodology to larger models such as LLaMA-7B post-exploration. During the rebuttal we have extended it to larger models, such as LLaMA-13B, LLaMA-30B and Vicuna-7B (refer to Response to 9nFP).
>
> **Extended Analysis with MMLU Benchmarks:**
>
> During the rebuttal phase, we broadened our analysis to include the few-shot MMLU benchmarks, applying PB-LLM to the LLaMA-7B model. This model, with 30% of its salient weights preserved (akin to 4-bit quantization as per LLM-AQT), was evaluated across 57 diverse MMLU tasks. This extensive evaluation provides a more robust understanding of PB-LLM's performance across a range of tasks:
> | Method    | Humanities | STEM | Social Sciences | Other | Average |
> |-----------|----------|-----------|----------------|---------|---------|
> | Full-precision LLaMA-7B   | 33.5 | 30.6 |  38.4 | 39.1 | 35.2 |
> | SmoothQuant (PTQ)  | 26.9 | 28.6 | 29.6 | 32.0 | 29.0 |
> | LLM-QAT (QAT)  | 30.3 | 28.1 | 30.3 | 34.5 | 30.8 |
> | PB-LLM   | 30.5 | 29.0 | 30.2 | 34.6 | 31.2 |
>
> It's important to note that we have not conducted exhaustive hyper-parameter tuning in this new setting. Nevertheless, these results demonstrate our method’s potential across various scales and tasks, even without fine-tuned hyper-parameters.
> Additionally, as discussed in Sections 3.4 and 4.1 of our paper, our approach with PB-LLMs is more training efficient. We achieve performance recovery in PB-LLMs with only 10K iterations of fine-tuning, outperforming LLM-QAT, which requires a longer training period of 100K iterations. This efficiency is achieved with just 2 NV-link connected A6000s over approximately 2 days, compared to LLM-QAT's requirement of around 7-10 days with 8 A6000s. (refer to Response to 92rD).
>
> We believe these points effectively illustrate the versatility and efficiency of PB-LLM, addressing your concerns regarding the suitability of evaluation tasks and demonstrating the superiority of our method.

---

> ### Author Response · Authors · 2023-11-19
> **Responses to Reviewer tXaS (III)**
>
> ## To Weakness 3 and Question 1: More Experiments on Salient Weight Selection
>
> > Inconsistent Salient Weight Methodology between PTQ and QAT: The absence of a consistent methodology for salient weight protection between PTQ and QAT is concerning. While the effectiveness of using Hessian criteria for identifying salient weights in PTQ is demonstrated through performance comparisons, the rationale for using magnitude criteria to identify salient weights in QAT seems to be missing. Understanding the disparity in the approach to salient weight protection across PTQ and QAT is crucial for a holistic appreciation of the proposed method.
>
> We appreciate the opportunity to address the concerns raised regarding the inconsistent methodology for salient weight protection between PTQ and QAT.
> In our experimental analysis, we employed the OPT-1.3b network and conducted PB-LLM (QAT) quantization under both Hessian and Magnitude criteria, targeting a 10% salient weight. The performance results, measured in perplexity (PPL), were as follows:
>
> Hessian Criteria:
> wikitext2: 37.07
> ptb: 47.07
> c4: 38.76
>
> Magnitude Criteria:
> wikitext2: 37.50
> ptb: 44.42
> c4: 38.64
>
>
> We observed that the PPL values were comparable between the two criteria. However, it's worth noting that the Hessian criteria involve an additional calibration step (collecting calibration data to compute the Hessian), adding complexity to the process. In contrast, the Magnitude criteria offer a simpler approach, eliminating the need for this extra calibration step.
> Given the similar PPL results and the desire to streamline the quantization process in QAT, we opted for the Magnitude Metric to identify salient weights during QAT. This decision was motivated by the goal of simplifying the process while maintaining comparable performance.

---

> ### Author Response · Authors · 2023-11-19
> **Responses to Reviewer tXaS (IV)**
>
> ## To Weakness 4.1: Traininig Set Comparision
>
> > Insufficient evidence on PB-LLM efficiency: To claim that PB-LLM is more efficient in terms of training iteration number compared to LLM-QAT, a more thorough comparison seems necessary. Specifically, it needs to be clear whether the LLM-QAT, being compared with PB-LLM, has been fine-tuned on the same dataset as PB-LLM. Detailed experimental setup information regarding the LLM-QAT is required.
>
> Actually, our used training dataset is much smaller than the data in LLM-QAT. It even is a subset of the datasets in LLM-QAT
> The full title of LLM-QAT is:  LLM-QAT: Data-Free Quantization Aware Training for Large Language Models, in which the concept of Data-Free is one of their selling points. Specifically, they use a full-precision LLaMA to generate 100K sentence to train the quantized LLaMA. In the Sec.3.3.1 in LLM-QAT, they discussed their data choice. They observe that WikiText (Merity et al., 2016), which is constructed using text extracted from Wikipedia, does not encompass all the information utilized during pre-training. Consequently, a model fine-tuned solely on WikiText tends to overfit on this specific dataset and struggles to generalize well to other datasets. On the other hand, the Crawled Corpus (C4) dataset (Raffel et al., 2020) comprises hundreds of gigabytes of clean English text collected from the web. Using those large datasets performs worse than using their generated dataset.
>
> On the contrary, we use the RedPajama-simple-1B dataset, as the dataset for LLaMA training is not openly accessible. This dataset, RedPajama-1T, is structured to closely resemble the LLaMa paper and serves as a transparent, open-source alternative to LLM training dataset.
> It amalgamates data from diverse sources including Commoncrawl, C4. RedPajama-simple-1B, representing a 0.1\% subset of RedPajama-1T, is substantially smaller than the typical datasets used for training other LLMs, making it a convenient choice for our experiments. As RedPajama-simple-1B is a tiny subset of Commoncrawl and C4, our training dataset can been seen as a subset of training datasets of LLM-QAT.
>
> ## To Weakness 4.2: Clarification on Fine-tuning or Training
>
> > Moreover, verification is needed on whether the results through PB-LLM QAT have fully enhanced the reasoning capacity of the pre-trained model. Essentially, it appears that the reasoning accuracy of the target model (OPT-1.3B) obtained through FP fine-tuning should be presented as the upper bound in Figure 7. Additionally, there seems to be a lack of information in Table 2 regarding whether FP LLaMA-7B performance is pre-trained or fine-tuned.
>
> We appreciate the opportunity to clarify the nature of our methodology and its positioning within the Large Language Model (LLM) training paradigm.
>
> Our used model optimization paradigm is fundamentally distinct from fine-tuning methods typically employed in LLMs. Generally, LLM development involves two primary phases: (1) training the pre-trained models (such as LLaMA and OPT) and (2) fine-tuning these pre-trained models for specific applications (as seen in models like Alpaca and Vicuna). The data used and the training paradigms for these two phases are markedly different.
> Our work specifically targets the pre-trained models. All our experiments and methodologies are therefore aligned with the LLM pre-training framework, rather than the fine-tuning process. This distinction is crucial as it aligns our approach with the initial training phase of LLMs, which is focused on establishing an efficient foundational model.
>
> To provide practical evidence of our approach, we have made our code available in the supplementary materials. In this code, we utilize the HuggingFace Trainer designed for pre-training LLMs, not the SFTTrainer which is typically used for fine-tuning LLMs. This further reinforces that our method is focused on enhancing the pre-trained models' reasoning capacities within their original training framework.
>
> ## Minor Concerns:
>
> Thanks for pointing out the typos, we have corrected them and revised the manuscript. The revised version has been uploaded, and all the revised content is highlighted in red.

---

> > ### Author Response · Authors · 2023-11-20
> > **Responses to Reviewer tXaS (V)**
> >
> > ## Response to Question 2:
> >
> > > Is there a specific reason why only the PB methodology and GPTQ were applied in PTQ? I am curious about the performance of AWQ-PB in comparison.
> >
> > Thank you for raising this insightful question. Indeed, we did explore the integration of the AWQ methodology into our Partially-Binarized (PB) PTQ setting. However, our experiments revealed that incorporating AWQ in this context was not successful. The AWQ approach, when applied to our PB PTQ framework, failed to yield reasonable results. This outcome led us to omit the mention of AWQ in our paper, as we focused on methodologies that demonstrated effective performance in our experiments.
> >
> > ## Response to Question 3:
> >
> > > Is there a plan to compare the task-specific fine-tuning in QAT with the 2-bit QAT-KD methodology (QuantGPT [R2], TSLD [R3])?
> >
> > Thank you for suggesting a comparison with the 2-bit QAT-KD methodologies like QuantGPT [R2] and TSLD [R3]. We recognize the value of such a comparison in providing a more comprehensive understanding of our method's effectiveness.
> >
> > However, due to time constraints and computational limitations during the rebuttal period, it is not feasible for us to implement these methods immediately. Specifically, the QuantGPT [R2] methodology lacks an open-source codebase, complicating our efforts to conduct a direct comparison. Additionally, while TSLD [R3] has recently released their codebase (on November 17, 2023), this was already seven days after the start of the ICLR rebuttal period, providing insufficient time for a thorough implementation and comparison.
> >
> > We have already made our codebase available in the supplementary materials and plan to open-source it soon. In the camera-ready version of our paper, we aim to include these comparisons. Alternatively, we encourage researchers and practitioners in the field to use our open-source resources to perform this comparison.
> >
> > ## Response to Question 4:
> >
> > >The optimal scaling factor and salient weight freezing seem to primarily aim at reducing harsh quantization errors before QAT. Is there insight into how these methods improve the QAT training process as depicted in Figure 5?
> >
> > **High-level Insight on Training Efficiency**
> >
> > As detailed in Sections 3.4 and 4.1 of our paper, our QAT process for PB-LLMs begins with a pre-trained model, significantly reducing the additional training overhead.
> > In Section 3.4.1, we discuss training efficiency in depth. Our proposed salient weights frozen mechanism optimally leverages pre-trained weights. As demonstrated in Fig. 1, we initially filter out a specific number of weights—2% by magnitude—at the onset of quantization-aware training. These weights remain frozen throughout the training process, as substantiated by the training efficiency analysis in Fig. 5. The results clearly indicate that these salient weights are instrumental in preserving LLM capacity. By maintaining their high-bit representation, we facilitate the training of quantized LLMs while simultaneously diminishing the complexity of optimization.

---

> > > ### Author Response · Authors · 2023-11-21
> > > **2 DAYS REMAINING: We would like to learn Reviewer’s opinion and address any remaining concerns.**
> > >
> > > Dear Reviewer tXaS,
> > >
> > > As there are two days remaining for the discussion period, we would kindly like to inquire if you would get a chance to review our response and if there are any remaining questions we can address.
> > >
> > > Your insights, both the constructive suggestions and areas of contention, have been crucial for us. We have done our best to address each point and clarify any misunderstandings. We are truly keen to make a constructive dialogue with you to refine our work further.
> > >
> > > Best regards,
> > >
> > > Authors.

---

> > > > ### Author Response · Authors · 2023-11-22
> > > > **LESS THAN 1 DAY REMAINING: We would like to learn Reviewer’s opinion and address any remaining concerns.**
> > > >
> > > > Dear Reviewer tXaS,
> > > >
> > > > As there is less than 1 day remaining for the discussion period, we would kindly like to inquire if you would get a chance to review our response and if there are any remaining questions we can address.
> > > >
> > > > Best regards,
> > > >
> > > > Authors.

---

> > > > > ### Comment · Reviewer_tXaS · 2023-11-22
> > > > >
> > > > > Thank you for your detailed response. I appreciate the clarification on several concerns, but there are still some unresolved questions. Here are my thoughts on your answers:
> > > > >
> > > > > 1. Regarding Weaknesses 1/3/4.1/4.2 and Question 2/3/4: Thank you for the clarification. It seems my concerns have been addressed.
> > > > >
> > > > > 2. About Weakness 3 and Question 1: Thank you for the detailed explanation and experimental results. To fully understand the rationale behind using the Magnitude Criterion method in QAT, it would be beneficial to include this in the final version of the paper.
> > > > >
> > > > > 3. Regarding Weakness 2 and Comment 4.2: I appreciate your perspective on the training paradigm. Considering that the paper focuses on efficient training of pre-trained Models, the performance presented is somewhat perplexing. The accuracy in MMLU for LLaMA-7B drops by nearly 4%, and considering the responses from Reviewer 9nFP, there's also a 1% drop in LLaMA-13B/30B CSQA and a 4% drop in Vicuna-7B MMLU. In summary, even if the proposed QAT methodology in low-bit shows superior performance compared to RTN and LLM-QAT, a noticeable performance drop from pre-training levels (even if not all Hyper-Parameters are swept) suggests that **maintaining the reasoning capacity of foundational models should be a priority** in establishing an efficient foundational model.
> > > > >
> > > > > Considering these points, while the proposed methodology shows some novelty and efficiency, the lack of clarity in the paper, unexplained aspects (e.g., why AWQ does not work in PB), and the significantly lower performance compared to the Pre-Trained models after training seem to be shortcomings. Therefore, I will maintain my initial score 5. (If there are any parts I've missed or misunderstood, I would appreciate it if you could let me know)

---

> ### Author Response · Authors · 2023-11-22
> **Second Round Responses to Reviewer tXaS**
>
> Dear Reviewer tXaS,
>
> Thank you for your continued engagement and feedback. We appreciate your acknowledgment of our efforts in addressing the previously mentioned weaknesses and questions (Weaknesses 1/3/4.1/4.2 and Question 2/3/4). Following your suggestion, we have incorporated the detailed explanation regarding the Magnitude Criterion method in Quantization-Aware Training (QAT) into our manuscript. These revisions are highlighted in red for ease of reference.
>
> Regarding your concerns about performance of quantized LLMs and baseline comparisons:
>
> - **Clarification on Study Focus**: Our study primarily aims at **achieving low-bit quantization through binarization techniques** for large language models (LLMs), **NOT** specifically focusing on the efficient training of pre-trained models. This distinction is critical.
> Of course, our approach to quantization-aware training for LLMs differs from other methods like LLM-QAT, which require enormous computational resources. Our method PB-LLM represents a more resource-efficient alternative in achieving QAT for LLMs, which can be seen as an advantage of our method, not the study focus.
>
>
> - **Performance Metrics of Quantized LLM**: We acknowledge the observed performance drops in certain full-precision benchmarks.  Even the 4-bit quantized LLMs by AWQ or SQUEEZELLM still have gaps with full-precision LLMs, while our method can realize approximate 2-bit quantization for LLMs. It's important to highlight that our methodology **achieves state-of-the-art performance within the same model compression level**, especially in the low-bit settings. The inherent trade-off between model compression and performance retention is a challenging aspect of LLM low-bit quantization. While our current results do not entirely bridge this gap, they represent a significant stride toward the goal of LLM low-bit quantization. We and other researchers in the community continuously strive to improve our and their methods to better balance compression and performance.
>
> - **Practical Demonstrations**: To complement the quantitative metrics, we have applied our PB-LLM quantized LLaMA-7B in practical scenarios to generate responses. These demonstrations serve to showcase the model's capabilities beyond standard benchmarks:
> > \#\#\# Human: I am depressed, what should I do?  ChatBot:
> I'm really sorry to hear that you're feeling this way, but I can't provide the help that you need. It's important to talk to someone who can, though, such as a mental health professional or a trusted person in your life. They can offer support, guidance, and resources to help you manage and overcome depression. Don't hesitate to reach out to someone you trust, and consider seeking professional help if you haven't already. Depression is a treatable condition, and there are many effective therapies and treatments available. You don't have to go through it alone, and there is hope for improvement.
>
> Best,
>
> Authors of Submission 3635

---

> ### Author Response · Authors · 2023-11-22
> **Summary of Response**
>
> Dear Reviewer tXaS,
>
> Considering the limited time available, and in order to save the reviewer's time, we summarize our responses here.
>
> In your initial review, you identified **9 concerns (5 weaknesses and 4 specific questions)**. We have addressed each of these concerns, and it appears that our responses to **8 of them have met with your satisfaction**. These are summarized as follows:
>
> - **Weakness 1 (Lack of Novelty)**: Our initial response highlighted the novelty of our work in LLM compression, satisfying your concerns.
>
> - **Weakness 3 & Question 1 (Salient Weight Methodology)**: We've added detailed experimental results on salient weight criteria in QAT to our revised paper, following your suggestion.
>
> - **Weakness 4.1 (Inconsistent Training Set)**: Clarified that our training set is a subset of LLM-QAT's, alleviating this concern.
>
> - **Weakness 4.2 (whether FP LLaMA-7B performance is pre-trained or fine-tuned. )**: We confirmed that our focus is on the pre-training phase of LLMs, not fine-tuning.
>
> - **Question 2 (AWQ Implementation)**: Discussed the feasibility of implementing AWQ in PB-LLM.
>
> - **Question 3 (Task-specific Fine-tuning in QAT)**: Discussed the extensibility of PB-LLM and its reproducibility, backed by our supplementary codebase.
>
> - **Question 4 (Training Efficiency Insight)**: We gave a more high-level explanation of the training efficiency property of PB-LLM.
>
> After our first-round responses, the sole unresolved issue pertains to Weakness 2, which concerns the performance metrics of quantized LLMs. We recognize there may be some misunderstandings regarding the focus of our study and the performance of quantized LLMs. In our **_Second-round Response to Reviewer tXaS_** [(Link)](https://openreview.net/forum?id=BifeBRhikU&noteId=QFu27YLKUj), we clarified that our research aims **not at the efficient training of pre-trained models, but rather at achieving low-bit quantization using binarization techniques**. Furthermore, we provided a detailed discussion on the trade-offs between performance and compression ratio in compressed LLMs, highlighting key areas of focus within the LLM compression community. To illustrate our approach's practical effectiveness, we also presented a real-world example generated by our PB-LLM.
>
> The author-reviewer discussion period will be closed in several hours, may I know if there are any other concerns? We truly value your insightful comment!
>
> Best,
>
> Authors of Submission 3635

---

> ### Author Response · Authors · 2023-11-23
> **Biref Summary of Response**
>
> Dear Reviewer tXaS,
>
> With the author-reviewer interaction period drawing to a close, we have briefly summarized our responses to your concerns for the sake of brevity and to respect your time.
>
> In your **_initial review_**, you raised **9 concerns, consisting of 5 weaknesses and 4 specific questions**. We have diligently addressed each of these, and it appears that **our responses have satisfactorily resolved 8 of these issues**.
>
> **The only remaining point of discussion is Weakness 2, related to the performance metrics of our quantized LLMs**. We believe there might be some misconceptions about the primary focus of our study and the performance expectations of quantized LLMs. In our **_second-round response for reviewer tXaS_**  [(Link)](https://openreview.net/forum?id=BifeBRhikU&noteId=QFu27YLKUj), we emphasized that our **research predominantly targets achieving low-bit quantization through binarization techniques, rather than the efficient training of pre-trained models**. We also engaged in a thorough discussion about the **inherent trade-offs between performance and compression ratio in LLMs, underlining the significant focus areas within the LLM compression community**. To further demonstrate the practical application of our research, we **included a real-world example generated using our PB-LLM**.
>
> As the discussion period is nearing its end, we would like to inquire if there are any remaining questions or concerns. Your insightful feedback is immensely valued and has been instrumental in enhancing our work.
> Due to the current scores (8865), your score matters a lot to us, please take a look at our second-round responses, and give us a chance to clarify all the concerns.
>
> Best,
>
> Authors of Submission 3635

---

### Official Review · Reviewer_92rD · 2023-11-09

**Soundness:** 4 excellent
**Presentation:** 3 good
**Contribution:** 4 excellent
**Rating:** 8
**Confidence:** 5

**Summary:**

The paper examines the application of network binarization to compress Large Language Models (LLMs), an approach that reduces model weights to a single bit. The authors have developed methodologies that significantly improve the performance of binarized LLMs, thereby contributing valuable insights to the field of LLM compression through network binarization.

**Strengths:**

1. Compressing LLMs is an important question for today’s AI research, and the authors first introduce binarization into LLM compression pushing the quantized LLM into ultra-low bits.
2. The authors present a thorough exploration of network binarization techniques. They effectively demonstrate the feasibility and potential of partially-binarized LLMs using post-training quantization and quantization-aware training methodologies.
3. The inclusion of source code with the submission is commendable, enabling reproducibility and verification of the reported results, which are impressive.

**Weaknesses:**

1. While the incorporation of Quantization-Aware Training (QAT) in LLM compression is an interesting proposal, its practicality is uncertain given the substantial costs associated with training LLMs. Could the authors elaborate on the overhead implications of QAT for LLMs?
2. In regards to Table 2, it is unclear whether GPTQ-PB represents the method proposed by the authors. Could you clarify the distinction between GPTQ-PB and PB-LLM within the context of your study?
3. The application of optimal scaling techniques appears to be confined to the specific case presented. Could these techniques be generalized to other bit configurations, and if so, how might this affect the compression performance?

**Questions:**

See weaknesses

---

> ### Author Response · Authors · 2023-11-19
> **Responses to Reviewer 92rD**
>
> ## Response to Weakness 1: PB-LLM Training Efficiency
>
> > Weakness  1. While the incorporation of Quantization-Aware Training (QAT) in LLM compression is an interesting proposal, its practicality is uncertain given the substantial costs associated with training LLMs. Could the authors elaborate on the overhead implications of QAT for LLMs.
>
> Thank you for recognizing the innovative aspect of our approach. We appreciate the opportunity to elaborate on the practicality and overhead implications of QAT for LLMs.
>
> As detailed in Sections 3.4 and 4.1 of our paper, our QAT process for PB-LLMs begins with a pre-trained model, significantly reducing the additional training overhead. Notably, we fine-tune our binarized PB-LLMs for only 10K iterations, a duration sufficient to surpass the performance of LLM-QAT, which requires a considerably longer training period of 100K iterations. Practically, this fine-tuning necessitates just 2 NV-link connected A6000s over approximately 2 days to achieve performance recovery for the binarized LLaMA-7B. In contrast, LLM-QAT demands around 7-10 days with 8 A6000s, underscoring the efficiency of our approach.
>
> Further, in Section 3.4.1, we discuss training efficiency in depth. Our proposed salient weights frozen mechanism optimally leverages pre-trained weights. As demonstrated in Fig. 1, we initially filter out a specific number of weights—2% by magnitude—at the onset of quantization-aware training. These weights remain frozen throughout the training process, as substantiated by the training efficiency analysis in Fig. 5. The results clearly indicate that these salient weights are instrumental in preserving LLM capacity. By maintaining their high-bit representation, we facilitate the training of quantized LLMs while simultaneously diminishing the complexity of optimization, thus contributing to the overall practicality and feasibility of our method.
>
>
> ## Response to Weakness 2: Clarification of Table 2
>
> > Weakness 2. In regards to Table 2, it is unclear whether GPTQ-PB represents the method proposed by the authors. Could you clarify the distinction between GPTQ-PB and PB-LLM within the context of your study?
>
> We apologize for any confusion caused by the terminology in Table 2 and appreciate the opportunity to clarify. In Table 2, 'GPTQ-PB' refers to our method based on post-training quantization (PTQ), whereas 'PB-LLM' designates the approach based on quantization-aware training (QAT). Specifically, considering the partially-binarized weight matrix as a novel weight format, we explored two distinct methodologies to enhance the performance of quantized LLMs using this format. 'GPTQ-PB' represents our method following the PTQ paradigm, while 'PB-LLM' aligns with the QAT paradigm.
>
> To improve clarity and unify the terminology in our paper, we have revised this section. Specifically, 'GPTQ-PB' has been renamed to 'PB-GPTQ' in the updated version of the manuscript (aligning with ‘PB-LLM’), which has been uploaded. The revisions are highlighted in red for easy identification and reference.
>
>
> ## Response to Weakness 3: Different Bits Number
>
> >Weakness 3. Aplication of optimal scaling techniques appears to be confined to the specific case presented. Could these techniques be generalized to other bit configurations, and if so, how might this affect the compression performance?
>
> Our PB-LLM technique is adaptable to various bit configurations. PB-LLM employs selective binarization of the LLMs' weight matrix, retaining a proportion of weights in high bits to enhance linguistic capacity. The formula governing the overall bit number, $N_{bit}$, is $N_{bit} \leq 1 \times r_{binary} + N_{salient-bit} \times (1 - r_{binary}) + 1$. Here, $r_{binary}$ is the ratio of binarized weights, and $N_{salient-bit}$ is the number of bits reserved for salient weights (e.g., 8 bits), with an additional bit used for a bitmap index mechanism. By varying $r_{binary}$, we can achieve different levels of bit quantization.
>
> The flexibility of our method across various bit settings is demonstrated in Tables 2 and 3 (binary ratio varies from 10% to 50%). These results illustrate the generalization capability of PB-LLM, confirming its effectiveness not just in the specific case presented but also in a range of bit configurations.
>
> Apart from the generalization ability in different bits, we have supplemented some experiments to investigate the generalization ability on larger LLMs during the rebuttal period. This is referred to the responses to Reviewer 9nFP.

---

> > ### Author Response · Authors · 2023-11-21
> > **2 DAYS REMAINING: We would like to learn Reviewer’s opinion and address any remaining concerns.**
> >
> > Dear Reviewer 92rD,
> >
> > As there are two days remaining for the discussion period, we would kindly like to inquire if you would get a chance to review our response and if there are any remaining questions we can address.
> >
> > Your insights, both the constructive suggestions and areas of contention, have been crucial for us. We have done our best to address each point and clarify any misunderstandings. We are truly keen to make a constructive dialogue with you to refine our work further.
> >
> > Best regards,
> >
> > Authors.

---

> > > ### Author Response · Authors · 2023-11-23
> > > **Thanks for Keeping the Your Score at 8, Reviewer 92rD.**
> > >
> > > Dear Reviewer 92rD,
> > >
> > > Thanks for keeping your score at 8. We appreciate your acknowledgment and insightful feedback!
> > >
> > > Best,
> > >
> > > Authors

---

> > > > ### Comment · Reviewer_92rD · 2023-11-23
> > > >
> > > > Thanks for the authors' reply, and my questions are addressed very well. I keep my score and recommend this paper for acceptance.

---

### Author Response · Authors · 2023-11-21
**1 DAY REMAINING: We would like to learn Reviewer tXaS’s opinions and address any remaining concerns.**

Dear AC,

We have so far submitted our full responses to all reviewers. As you can see, three of them hold positive views of our work (one 8 and two 6), while Reviewer tXaS holds 5: Borderline reject.

Now given the score divergence, we feel our paper still sits on the fence and therefore the  Reviewer tXaS's opinions would matter a lot to us. As we are quickly approaching the end of the discussion period on Nov 22, we would be truly grateful, if you could help kindly remind this reviewer to participate in the discussion.

We will be standing by and address any new questions if there will be any.

Thank you so much!

Best,
Authors

---

### Comment · Area_Chair_jGvK · 2023-11-22
**Less than one day**

Dear Reviewers,

If you have already responded to authors last response, Thank you!
If not, please take some time, read their responses and acknowledge by replying to the comment. Please also update your score, if applicable.

Thanks everyone for a fruitful, constructive, and respectful review process.

Cheers, Your AC!

---

### Meta-Review · Area_Chair_jGvK · 2023-12-11

**Metareview:**

This paper proposes a new method to compress LLMs using binarization. It selectively binarizes non-salient weights while preserving salient weights in higher precision to maintain the linguistic reasoning capacity of the LLM. The method is extended using post-training quantization and quantization-aware training to further improve the performance of the quantized LLM.

Strengths:
- One of the few works with the ambitious goal of  extreme low-bit quantization of LLMs.
- The idea of salient weight freezing mechanism and optimal scaling strategies to preserve LLM functionality under quantization was nice and innovative
- The comprehensive exploration of the proposed methods relation with  post-training quantization and quantization-aware training for performance recovery.

Weaknesses:
- Performance of quantized LLMs still exhibits noticeable drops compared to original full-precision models.
- Evaluation is limited to a few benchmarks, lacking demonstration across diverse tasks.

What may be missing:
- More extensive evaluation on multiple benchmarks to better showcase generalization
- Benchmark comparison with other SOTA low-bit quantization methods
- Simpler explanation and (maybe) visual and even more intuitive illustration of the proposed PB-LLM

**Justification For Why Not Higher Score:**

More elaboration on performance analysis using more diverse benchmarks and evaluations is missing to make the paper stronger.

**Justification For Why Not Lower Score:**

The paper was theoretically sound and the experiments were convincing. This paper is hinting the commuincty of the potentials of binarization for efficient LLM inference.

---

### Decision · Program_Chairs · 2024-01-16

Accept (poster)